# Small RNA, Transcriptome and Degradome Analysis of the Transgenerational Heat Stress Response Network in Durum Wheat

**DOI:** 10.3390/ijms22115532

**Published:** 2021-05-24

**Authors:** Haipei Liu, Amanda J. Able, Jason A. Able

**Affiliations:** Waite Research Institute, School of Agriculture, Food & Wine, The University of Adelaide, Urrbrae, SA 5064, Australia; amanda.able@adelaide.edu.au (A.J.A.); jason.able@adelaide.edu.au (J.A.A.)

**Keywords:** heat stress, transgenerational stress memory, next-generation sequencing, microRNAs, cereal reproduction, crop improvement, tetraploid wheat

## Abstract

Heat stress is a major limiting factor of grain yield and quality in crops. Abiotic stresses have a transgenerational impact and the mechanistic basis is associated with epigenetic regulation. The current study presents the first systematic analysis of the transgenerational effects of post-anthesis heat stress in tetraploid wheat. Leaf physiological traits, harvest components and grain quality traits were characterized under the impact of parental and progeny heat stress. The parental heat stress treatment had a positive influence on the offspring for traits including chlorophyll content, grain weight, grain number and grain total starch content. Integrated sequencing analysis of the small RNAome, mRNA transcriptome and degradome provided the first description of the molecular networks mediating heat stress adaptation under transgenerational influence. The expression profile of 1771 microRNAs (733 being novel) and 66,559 genes was provided, with differentially expressed microRNAs and genes characterized subject to the progeny treatment, parental treatment and tissue-type factors. Gene Ontology and KEGG pathway analysis of stress responsive microRNAs-mRNA modules provided further information on their functional roles in biological processes such as hormone homeostasis, signal transduction and protein stabilization. Our results provide new insights on the molecular basis of transgenerational heat stress adaptation, which can be used for improving thermo-tolerance in breeding.

## 1. Introduction

Durum wheat (*Triticum turgidum* ssp. *durum*) is a widely grown tetraploid wheat (genomes AABB) used for human consumption. Durum wheat represents the most cultivated cereal crop in the Mediterranean region, accounting for more than half of its wheat growing area [1]. As a staple food crop, grains of durum wheat are used in a wide range of end-products, such as pasta, couscous, freekeh, unleavened and leavened bread [2,3]. Compared with common hexaploid wheat, durum grains have several unique quality traits, such as superior protein content, high grain hardness and an amber-colored vitreous endosperm [3,4]. However, in Mediterranean regions, abiotic stresses such as drought and heat are significant threats to durum wheat yield, grain quality and the characteristics of end-products [1,5,6,7]. In the Australian wheat belt, high temperature often occurs in the field after anthesis during grain development, and increases in intensity until maturity [8,9].

For cereal crops, achieving high grain productivity and quality relies on successful reproductive development [10]. Cereal reproduction processes, flowering and grain development, are both extremely sensitive to environmental stress [10,11]. Heat stress at flowering significantly inhibit photosynthesis due to thermal damage of the chloroplast protein–pigment complexes, causing pollen abortion and dehiscence that leads to floret sterility, which ultimately reduces grain set and grain number [10,12,13]. During grain fill, the optimal temperature for cereal crops is approximately 21 °C [14]. High temperatures at this stage have direct negative impacts on starch biosynthesis, reductive-oxidation balance, dry matter accumulation and assimilate transport, which all affect metabolite accumulation in the grains, leading to reduced grain weight and changes in grain quality characteristics [8,10,11]. With global climate change, extreme weather events such as high temperature episodes are predicted to only increase in frequency, intensity and duration. Plant breeders and associated researchers are therefore constantly being challenged to develop superior crop varieties with improved stress tolerance, higher yield stability and enhanced grain quality.

In general, heat stress tolerance refers to the ability of crops to make adaptive changes in plant growth and achieve reproductive success under high field temperature conditions [11,15]. The mechanistic basis of heat stress tolerance includes phenotypic plasticity, physiological adaptations, genetic regulation, epigenetic modifications, induced production of protective metabolites [e.g., reactive oxygen species (ROS) scavengers and glutathione] and protective proteins [e.g., heat shock proteins (HSPs) and rubisco activases] [10,15,16,17]. A significant body of work has investigated the short-term heat stress response networks in cereal crops [18,19,20,21]. Moreover, in the last decade, emerging studies have provided increasing evidence on the phenomenon of plant stress memory, which provides a new means of building stress resilience in the crop varieties of “tomorrow” [22,23,24,25,26,27,28].

Stress memory refers to changes at the molecular, physiological, proteomic and metabolomic level after the initial stress occurs, which then enables the plants to be more tolerant or adaptable to recurring stress within the current life cycle, or even future generations [29,30,31]. Establishment of stress memory requires sophisticated yet coordinated changes in gene expression, epigenetic modification and metabolite biosynthesis, which ultimately improves the efficiency of plants to rapidly activate the most effective and suitable response pathways [29,30,31,32]. In certain cases, stress memory has transgenerational effects, which can be passed on to progeny from parents or even grandparents [22,24,25,26,27,33,34]. For example, in bread wheat, pre-anthesis and post-anthesis heat stress priming applied in the parents had a positive influence on the tolerance of the next generation to post-anthesis stress [28]. Heat primed progeny had better physiological performance (e.g., higher photosynthetic rate and antioxidative activities) and higher yield than non-primed progeny. The influence of transgenerational priming was also prominent on the transcriptomic and proteomic levels, where genes and proteins related to signal transduction, photosynthesis and energy production were up-regulated in the primed progeny, contributing to a higher level of thermo-tolerance [28].

Epigenetic systems also have significant roles in stress memory and priming-induced tolerance [23,29,32]. Upon abiotic stress signals and developmental cues, epigenetic modifications (such as DNA methylation, histone modification, chromatin remodeling and the activities of small non-coding RNAs) can rapidly induce dynamic changes that affect gene expression at the post-transcriptional level, and the effects can be short term, long-lasting or even transgenerational [23,29,32]. Particularly, plant small RNAs [namely microRNAs (miRNAs) and small interfering RNAs (siRNAs)] are core components of the epigenetic regulatory system [35]. miRNAs are widely distributed and highly conserved in the plant kingdom, throughout model plants and crop species [36,37,38]. They sit at the center of regulating plant development and stress responses, with their functions in modulating gene expression in a sequence-specific manner at the post-transcriptional level [39,40,41,42,43]. In general, stress-induced miRNAs repress the expression of their target genes that have negative roles in stress adaptation, while stress-reduced miRNAs allow for the accumulation and increased activity of the target genes with positive functions [35,44,45,46,47]. Since the discovery of small RNAs, numerous studies have identified and characterized the functions of different miRNA families that are responsive to various stresses [48,49,50,51,52,53,54,55,56,57]. Only recently, emerging findings indicate that miRNAs may also hold the key as epigenetic regulators of intra- and trans-generational stress memory [24,25,26,58,59]. Our previous research has demonstrated that progeny of water-deficit and heat stress primed parents showed genotypic differences in germination traits, seedling vigor and adaptation to nitrogen starvation stress [24,26]. Previous research has also demonstrated that transgenerational exposure of a single water-deficit stress can help a less tolerant durum wheat genotype to cope better with reoccurring water deficiency in terms of physiological adaptation and yield performance, with contributions from miRNA-mRNA regulatory modules related to hormone signaling and cellular water relations [25]. In *Brassica rapa*, progeny of heat-stressed parents exhibited significant abundance changes in non-coding RNA fragments including tRNA fragments and small nucleolar RNA fragments, of which the targets showed a significant Gene Ontology (GO) enrichment in the cellular component endoplasmic reticulum and the molecular function of protein binding [60]. It was concluded that changes in the non-coding RNA population were inherited from the stressed parents; and such changes could contribute to the improvement of stress adaptation in future generations [60]. However, to date there has been no research on the parental heat stress priming in *Triticum turgidum*. Furthermore, the underlying physiological, genetic and epigenetic mechanisms of transgenerational heat stress adaptation remain largely unknown.

The purpose of the present study was to investigate the effects of parental and progeny post-anthesis heat stress on the leaf physiological traits, morphological traits, harvest components and grain quality characteristics in Australian durum wheat. To further explore the molecular basis of the transgenerational effects of heat stress, integrated next-generational sequencing analyses of the miRNAome, mRNA transcriptome and mRNA degradome were performed to identify regulatory miRNAs and genes associated with durum wheat transgenerational stress adaptation.

## 2. Results

### 2.1. Physiological, Grain Quality and Yield Performance of Two Durum Wheat Varieties under the Effects of Transgenerational and Progeny Heat Stress

Two high-yielding Australian durum wheat varieties, DBA Aurora and DBA Artemis, were used in the current study. DBA Artemis has a higher level of heat stress tolerance than DBA Aurora, while DBA Aurora has a higher level of water-deficit stress tolerance than DBA Artemis [8]. To investigate the transgenerational effects of heat stress, the seeds of the two varieties were collected from the control group parents and post-anthesis heat stress group parents from a previous study [8]. Four seed groups were used: AuC (DBA Aurora, seeds from control group parents), AuH (DBA Aurora, seeds from heat stress group parents), AtC (DBA Artemis, seeds from control group parents) and AtH (DBA Artemis, seeds from heat stress group parents). To investigate the effects of current-generation heat stress, each seed group was grown and treated with either the control condition or the heat stress condition. In total, the current study had eight treatment groups: AuCC (DBA Aurora progeny treated with the control, originated from control group parents), AuCH (DBA Aurora progeny treated with heat stress, originated from control group parents), AuHC (DBA Aurora progeny treated with the control, originated from heat stress group parents), AuHH (DBA Aurora progeny treated with heat stress, originated from heat stress group parents), AtCC (DBA Artemis progeny treated with the control, originated from control group parents), AtCH (DBA Artemis progeny treated with heat stress, originated from control group parents), AtHC (DBA Artemis progeny treated with the control, originated from heat stress group parents), AtHH (DBA Artemis progeny treated with heat stress, originated from heat stress group parents).

To evaluate leaf physiological response to heat stress (HS), flag leaf chlorophyll content and adaxial stomatal conductance (Figure 1) were measured every five days from anthesis to 40 days post-anthesis (i.e., 0–40 DPA). For leaf chlorophyll content, DBA Aurora and DBA Artemis showed differential response patterns to the parental and progeny HS factors. In DBA Aurora (Figure 1), chlorophyll content showed significant reduction in response to progeny HS in treatment groups from the control parents (AuCH vs. AuCC), starting from the first HS event (5DPA) to 40DPA. From 10DPA, chlorophyll content was significantly reduced by progeny HS in groups from the stressed parents (i.e., AuHH vs. AuHC). It is worth noting that the chlorophyll content of AuHH (HS treated progeny from the HS parents) was significantly higher than AuCH (HS treated progeny from the control parents) from 10 to 40DPA, showing the beneficial effects of parental HS on the progeny stress response. In DBA Artemis (Figure 1), the progeny HS treatment significantly reduced the chlorophyll content for both control parent groups (AtCH vs. AtCC) and stress parent groups (AtHH vs. AtHC), starting from 15 to 40DPA. The stressed progeny from the stressed parents (AtHH) only showed significantly higher chlorophyll content than the stressed progeny from the control parents (AtCH) at 15DPA. For stomatal conductance, the two varieties have shown a similar trend. In both DBA Aurora and DBA Artemis, progeny groups from the control parents (AtCH vs. AtCC, AuCH vs. AuCC) and the progeny groups from the stressed parents (AtHH vs. AtHC, AuHH vs. AuHC) had shown significant reduction in stomatal conductance from the onset of heat stress (5DPA) to 40DPA (Figure 1).

For yield components, the two varieties have also exhibited different responses for the traits measured (Table 1). For grain weight per plant, DBA Artemis had a significantly reduced response to heat stress for both control parents’ progeny groups and stressed parents’ progeny groups (AtCH vs. AtCC, AtHH vs. AtHC). There was no significant difference between progeny groups with the same progeny treatment but originated from different parents (AtHC vs. AtCC, AtHH vs. AtCH). However, for DBA Aurora, although heat stress had significantly reduced the grain weight for both control parents’ progeny groups and stressed parents’ progeny groups (AuCH vs. AuCC, AuHH vs. AuHC), the stressed progeny from the stressed parents (AuHH) had significantly higher grain weight than the stressed progeny from the control parents (AuCH). The exact same genotype-dependent trend can also be observed for biomass. For grain number per plant, DBA Artemis only showed a significant reduction in response to heat stress in the control parents’ progeny groups (AtCH vs. AtCC). In DBA Aurora, heat stress had significantly reduced the grain number for both control parents’ progeny groups and stressed parents’ progeny groups (AuCH vs. AuCC, AuHH vs. AuHC), but AuHH had significantly higher grain number than AuCH. For harvest index, DBA Artemis only showed a significant reduction in response to heat stress in the control parents’ progeny groups (AtCH vs. AtCC). For DBA Aurora, harvest index was only significantly reduced under heat stress for stressed parents’ progeny groups (AuHH vs. AuHC), but the harvest index of AuHH and AuCH had no significant difference. Neither of the two varieties showed any significant difference across treatment groups for 1000-grain weight.

For morphological traits, neither of the two varieties showed any significant difference across treatment groups for plant height and tiller number (Table 2). For main spike length, the stressed progeny from the stressed parents had the highest main spike length in DBA Artemis, whereas DBA Aurora treatment groups did not exhibit any significant difference. When comparing grain quality characteristics, DBA Aurora and DBA Artemis exhibited the same response pattern for grain protein content (GPC), where GPC only significantly increased in response to progeny HS in the groups from the control parents (i.e., AtCH vs. AtCC, AuCH vs. AuCC). For grain total starch content (TSC), neither of the varieties exhibited a significant response to progeny HS in groups from the same parents. However, the TSC of DBA Artemis groups from the stressed parents (AtHC and AtHH) was significantly higher than AtCC. The TSC of the DBA Aurora groups from the stressed parents (AuHC and AuHH) was significantly higher than AuCH. Neither of the two varieties showed any significant difference across treatment groups for flour yellowness pigmentation (*b).

### 2.2. Analysis of the miRNAome and Differentially Expressed miRNAs (DEMs)

Flag leaf and developing grain samples of DBA Aurora were used for miRNAome, mRNA transcriptome and degradome sequencing analysis. DBA Aurora was chosen based on the positive impacts that the parental stress treatment had on progeny stress adaptation as observed in the physiological, morphological, yield and grain quality traits. A total of eight sRNA libraries (two tissue types by four treatment groups) were constructed and sequenced. Over 94.13 million raw reads were generated, with over 36.40 million being unique reads (Appendix A). After reads clean-up and filtering, over 71.25 million clean sRNA reads were obtained, of which nearly 31.56 million were unique sRNA reads. After bioinformatic analysis and reads annotation, 1771 MIR-miRNA entries (considering both different MIR gene origins and different mature miRNA products) were identified (Appendix A). The miRNA entries were categorized into five groups (group 1 to group 5), where group 1 to 4 contains different sub-categories of conserved miRNAs (a total of 1038 miRNAs), and group 5 contains all the novel miRNAs (733) identified in this study. The conservation profile (Figure 2a) demonstrates the number of conserved miRNAs shared between the current study and the reference plant species in the miRBase. The top three conserved species were bread wheat (*Triticum aestivum*), rice (*Oryza sativa*) and goat grass (*Aegilops tauschii*). *Triticum turgidum* ranked last due to the limited number of ttu-miRNAs registered in the miRBase. The distribution of miRNAs across different treatment groups are shown by Venn diagrams for both the flag leaf and the developing grains tissue (Figure 2b).

The normalized reads counts of conserved and novel miRNAs were used to identify significant DEMs (*p* < 0.05 and |log2foldchange| > 1) in pair-wise comparisons subject to different factors. To investigate how miRNAs respond to progeny HS treatment, pair-wise comparisons were made between treatment groups of the same parental origin: AuCH vs. AuCC and AuHH vs. AuHC (Appendix A). For treatment groups from the control parents, 561 miRNAs showed a significant stress responsive expression pattern (*p* < 0.05 and |log2foldchange| > 1) to the progeny HS treatment in flag leaf tissue (AuCH_L vs. AuCC_L), and 459 miRNAs showed a significant stress responsive expression pattern in the developing grains (AuCH_G vs. AuCC_G). For treatment groups from the heat stressed parents, 366 miRNAs showed a significant stress responsive expression pattern (*p* < 0.05 and |log2foldchange| > 1) to the progeny HS treatment in flag leaf tissue (AuHH_L vs. AuHC_L) and 460 miRNAs showed a significant stress responsive expression pattern in the developing grains (AuCH_G vs. AuCC_G).

To investigate how miRNA expression was affected by the parental treatment factor, pair-wise comparisons were made between the same progeny treatment groups with a different parental origin: AuHC vs. AuCC and AuHH vs. AuCH (Appendix A). For the control progeny treatment groups, 667 miRNAs showed a significant expression pattern (*p* < 0.05 and |log2foldchange| > 1) due to the different parental treatment in the flag leaf tissue (AuHC_L vs. AuCC_L) and 488 miRNAs showed a significant expression pattern in the developing grains (AuHC_G vs. AuCC_G). For the heat stressed progeny groups, 187 miRNAs showed a significant expression pattern (*p* < 0.05 and |log2foldchange| > 1) due to the different parental treatment in the flag leaf tissue (AuHH_L vs. AuCH_L) and 430 miRNAs showed a significant expression pattern in the developing grains (AuHH_G vs. AuCH_G).

To investigate miRNAs with a tissue-specific expression pattern, pair-wise comparisons were made between the flag leaf and developing grain samples of the same treatment group (Appendix A). For the control progeny treatment group from the control parents, 920 miRNAs showed significant tissue-specific expression (*p* < 0.05 and |log2foldchange| > 1) (AuCC_L vs. AuCC_G). For the heat stressed progeny group from the control parents, 1083 miRNAs showed significant tissue-specific expression (*p* < 0.05 and |log2foldchange| > 1) (AuCH_L vs. AuCH_G). For the control progeny treatment group from the stressed parents, 1087 miRNAs showed significant tissue-specific expression (*p* < 0.05 and |log2foldchange| > 1) (AuHC_L vs. AuHC_G). For the heat stressed progeny group from the stressed parents, 1067 miRNAs showed significant tissue-specific expression (*p* < 0.05 and |log2foldchange| > 1) (AuHH_L vs. AuHH_G).

### 2.3. Analysis of the mRNA Transcriptome, Degradome and Differentially Expressed Genes (DEGs)

Transcriptome sequencing of the eight libraries generated over 689 million reads, of which over 655 million can be mapped to the reference genome assembly (Appendix A). A total of 66,559 genes (Appendix A) have been identified from the sequencing reads, and their normalized abundance in each library was represented as FPKM (Fragments Per Kilobase Million). Significant DEGs (*p* < 0.05 and |log2foldchange| > 1) were identified in pair-wise comparisons subject to different factors.

To identify DEGs in response to the progeny HS treatment factor, pair-wise comparisons were made between treatment groups of the same parental origin: AuCH vs. AuCC and AuHH vs. AuHC (Appendix A). For treatment groups from the control parents, 2756 DEGs showed a significant stress responsive pattern (*p* < 0.05 and |log2foldchange| > 1) to the progeny HS treatment in the flag leaf tissue (AuCH_L vs. AuCC_L) and 263 DEGs showed a significant stress responsive pattern in the developing grains (AuCH_G vs. AuCC_G). For treatment groups from the heat stressed parents, 2378 DEGs showed a significant stress responsive pattern (*p* < 0.05 and |log2foldchange| > 1) to the progeny HS treatment in the flag leaf tissue (AuHH_L vs. AuHC_L) and 531 DEGs showed a significant stress responsive pattern in the developing grains (AuCH_G vs. AuCC_G). Interestingly, in all these comparisons, the number of DEGs that were up-regulated under progeny heat stress was always higher than the number of down-regulated DEGs (Figure 3a). KEGG pathway enrichment analysis of these progeny heat stress-responsive DEGs revealed the functional regulatory pathways that the genes are associated with (Figure 4). Some KEGG pathways were commonly enriched for progeny groups from the control and the stressed parents, while some KEGG pathways were specific to different pair-wise comparisons. For example, in the flag leaf, KEGG pathway ABC transporters (ko02010) were enriched for progeny groups from both the control and the stressed parents (AuCH_L vs. AuCC_L and AuHH_L vs. AuHC_L). Similarly, in the developing grains, MAPK signaling pathway—plant (ko04016) was also enriched for both AuCH_G vs. AuCC_G and AuHH_G vs. AuHC_G. However, in the flag leaf, KEGG pathways like phenylpropanoid biosynthesis (ko00940) and photosynthesis—antenna proteins (ko00196) were exclusively enriched for progeny groups from the stressed parents (AuHH_L vs. AuHC_L). In the developing grains, pathways such as flavonoid biosynthesis (ko00941) and brassinosteroid biosynthesis (ko00905) were also only enriched for progeny groups from the stressed parents (AuHH_G vs. AuHC_G).

To investigate how gene expression was affected by the parental treatment factor, pair-wise comparisons were made between the same progeny treatment groups with a different parental origin: AuHC vs. AuCC and AuHH vs. AuCH (Appendix A and Figure 3b). For control progeny treatment groups, 1133 DEGs showed a significant expression pattern (*p* < 0.05 and |log2foldchange| > 1) due to the different parental treatment in the flag leaf tissue (AuHC_L vs. AuCC_L) and 203 DEGs showed a significant expression pattern in the developing grains (AuHC_G vs. AuCC_G). For heat stressed progeny groups, 1705 DEGs showed a significant expression pattern (*p* < 0.05 and |log2foldchange| > 1) due to the different parental treatment in the flag leaf tissue (AuHH_L vs. AuCH_L) and 79 DEGs showed a significant expression pattern in the developing grains (AuHH_G vs. AuCH_G). KEGG pathway enrichment analysis of parental heat stress-responsive DEGs revealed the functional regulatory pathways that the genes are associated with (Appendix A). Some KEGG pathways were commonly enriched for progeny groups with different treatments, while some KEGG pathways were specific to different pair-wise comparisons. For example, in the flag leaf, ko00480 glutathione metabolism and ko00500 starch and sucrose metabolism were enriched for both control and stressed progeny groups under transgenerational effects (AuHC_L vs. AuCC_L and AuHH_L vs. AuCH_L). KEGG pathways like carotenoid biosynthesis (ko00906) and carbon fixation in photosynthetic organisms (ko00710) were only enriched for the control progeny groups (AuHC_L vs. AuCC_L), while anthocyanin biosynthesis (ko00942) and MAPK signaling pathway—plant (ko04016) were only enriched for the heat stressed progeny groups (AuHH_L vs. AuCH_L).

To identify DEGs with a tissue type-dependent pattern, pair-wise comparisons of gene expression were made between leaf and grain libraries from the same treatment group (Appendix A). In the AuCC group, 9155 DEGs had significant differential expression (*p* < 0.05, |log2foldchange| > 1) between the flag leaf and the grain libraries; in AuCH group, there were 9662 tissue-dependent DEGs; in AuHC group, there were 8870 tissue-dependent DEGs; and in AuCH group, there were 9473 tissue-dependent DEGs. In all these comparisons, the number of DEGs that were up-regulated (i.e., more abundant in the leaf tissue) was always higher than the number of down-regulated DEGs (i.e., more abundant in the developing grains) (Figure 3c).

Degradome sequencing of the eight mRNA degradome libraries generated over 608 million raw reads; among these, over 69 million were unique reads that can be mapped to the reference genome (Appendix A). The number of durum wheat reference transcripts that the reads were aligned to (i.e., number of covered transcripts) varied across the eight degradome libraries (ranging from 143,896 to 164,573). Further bioinformatics analysis profiled the miRNA-guided mRNA cleavage signatures within protein-coding gene transcripts. In total, 198,472 miRNA-targeted transcript sites were identified in the flag leaf libraries and 231,116 target sites were identified in the developing grain libraries (Appendix A). Appendix A also provides results of the detailed analysis of the degradomics (including transcript positions of miRNA:mRNA pairing, matched sequences between miRNA and mRNA targets and normalized degradome tags count in each library).

### 2.4. Identification of Key miRNA-mRNA Modules with Integrated Omics Analysis

Analysis integrating the three types of sequencing data (small RNAome, transcriptome and degradome) was conducted as previously described [25,48,49], to discover miRNA-mRNA pairs with significant expression that showed an antagonistic pattern, according to the mRNA silencing effect of miRNA (Appendix A). Briefly, validated pairs of miRNA and their targets were first selected where their expression (sRNA-seq and mRNA-seq) and pairing (degradome-seq) were confirmed. After which, only the pairs where both miRNA and mRNA showed a significant expression pattern were chosen. From these pairs, the ones where miRNA and mRNA had shown an antagonistic/opposite regulatory pattern were selected (i.e., significantly reduced miRNA expression matched to significantly increased mRNA expression, significantly increased miRNA expression matched to significantly decreased mRNA expression).

Subject to the progeny HS treatment factor, 1009 pairs of miRNA-mRNAs exhibited a significant antagonistic pattern in the groups from the control parents in the leaf tissue (AuCH_L vs. AuCC_L); and 322 pairs in the grain tissue (AuCH_G vs. AuCC_G). A total of 1108 pairs of miRNA-mRNAs exhibited a significant antagonistic pattern in the groups from the stressed parents in the leaf tissue (AuHH_L vs. AuHC_L); and 361 pairs in the grain tissue (AuHH_G vs. AuHC_G). Within the same tissue type, the number of significant miRNA-mRNA pairs were higher in the groups from the stressed parents compared with the ones from the control parents, demonstrating that transgenerational HS priming has elicited a higher degree of responsiveness of the miRNA-mRNA modules. Further classification analysis of KEGG pathways associated with these miRNA-mRNA modules revealed the percentage distribution of the key biological processes regulated in different pair-wise comparisons (Figure 5). The majority of the heat stress-responsive miRNA-mRNA modules participate in metabolism-related KEGG pathways, such as carbohydrate metabolism, which had the highest percentage enriched for all comparisons. Some differences can be observed for progeny groups from different parents. For example, in the flag leaf, transcription (under genetic information processing category) had a higher percentage in progeny groups from the stressed parents (AuHC_L vs. AuHH_L) than those from the control parents (AuCH_L vs. AuCC_L). In the developing grains, comparing the progeny groups from the stressed parents (AuHC_G vs. AuHH_G) to the ones from the control parents (AuCH_G vs. AuCC_G), a higher percentage of miRNA-mRNA pairs were involved in lipid metabolism but a lower percentage was found for transport and catabolism.

In response to the parental HS treatment, 715 pairs of miRNA-mRNAs exhibited an antagonistic expression pattern in the control treated progeny from different parents in the leaf tissue (AuHC_L vs. AuCC_L); and 440 pairs in the grain tissue (AuHC_G vs. AuCC_G). A total of 511 pairs of miRNA-mRNAs exhibited an antagonistic expression pattern in the HS treated progeny from different parents in the leaf tissue (AuHH_L vs. AuCH_L); and 182 pairs in the grain tissue (AuHH_G vs. AuCH_G).

While looking into the tissue-type dependent pattern, for progeny groups from the control parents, 5621 pairs of miRNA-mRNAs exhibited an antagonistic expression between AuCC_L vs. AuCC_G; and 7535 pairs between AuCH_L vs. AuCH_G. For progeny groups from the stressed parents, 7618 pairs of miRNA-mRNAs exhibited an antagonistic expression between AuHC_L vs. AuHC_G; and 7395 pairs between AuHH_L vs. AuHH_G. Under optimal conditions for the progeny (control), the parental HS treatment appeared to have elicited a greater degree of tissue-dependent pattern of miRNA-mRNA modules (7618 pairs vs. 5621 pairs).

### 2.5. qPCR Profiling of Stress-Responsive DEMs and DEGs

Seven stress-responsive miRNAs (Figure 6) and 11 target genes (Figure 7) were chosen for qPCR expression analysis, based on their biological functions associated with stress adaptation and plant development. Different expression patterns of stress responsive miRNAs and their target genes were observed, subject to different factors (progeny treatment, parental treatment and tissue-type). The seven miRNAs were expressed in both flag leaf and developing grain tissues (Figure 6). In the flag leaf tissue, the seven miRNAs all showed a significantly decreased expression pattern in response to the progeny heat stress treatment in the groups from the control parents. There was no significant difference between control and heat stressed progeny from the stressed parents for all miRNAs profiled, because their expression was already lowered (primed), showing no statistical difference to the stressed progeny from the control parents (except for bid-miR5054_L+2 where AuHH_L actually had even lower expression than AuCH_L). In the grains, only ttu-miR160 showed significant difference across four treatment groups where AuCH_G had the highest level of expression.

Some of the target genes showed a tissue-specific expression pattern (Figure 7). The auxin response factor 22 (target of ttu-miR160) was not expressed in the flag leaf samples, while the catalase-1 (target of bid-miR5054_L+2), bZIP transcription factor (target of tae-miR408_L-1), auxin response factor 17 (target of tae-miR408_L-1) and FKBP13 (target of ttu-miR160) were not expressed in the developing grains (Figure 7). Five genes, including catalase-1, Cu/Zn superoxide dismutase, bZIP transcription factor, auxin response factor 17 and FKBP13 (peptidyl-prolyl cis-trans isomerase FKBP13) did not show any detectable expression in the flag leaf tissue of the treatment group AuCC, suggesting that their expression was triggered by progeny heat stress and the transgenerational heat stress treatment.

Lower miRNA expression in the progeny groups from the stressed parents would allow for higher expression of their target genes that have positive functions in stress adaptation. For example, in the flag leaf tissue, several targets including the F-box protein (target of ata-miR528-5p), catalase-1 (target of bid-miR5054_L+2), Cu/Zn superoxide dismutase (Cu/Zn SOD, target of osa-miR398a_L+1R-1), phospholipid-transporting ATPase (target of osa-miR837), auxin response factor 17 (target of tae-miR408+L-1) and FKBP13 (target of ttu-miR160) all displayed high expression levels in the stressed progeny group from the stressed parents (AuHH_L), thus suggesting the positive priming effects of transgenerational stress. In the developing grains, however, the expression of miRNAs and targets was less dynamic. Only two target genes, Cu/Zn superoxide dismutase and auxin response factor 22, showed the highest expression in the stressed progeny group from the stress parents (AuHH_G), while the other target genes showed no significant difference across treatment groups (Figure 7).

## 3. Discussion

Heat stress has been a significant threat to cereal crop production globally, particularly in regions where unpredicted rising temperatures occur during flowering and grain development stages [10]. Historically, breeding programs have been focused on improving thermotolerance by selecting germplasm with adaptive traits. Emerging findings on intra-generational and transgenerational stress memory have provided new opportunities and strategies for fortifying stress resilience in crops [22,61]. Stress priming, also referred to as stress hardening or stress conditioning, utilizes the mechanisms of stress memory to elicit adaptive responses and long-lasting changes in plants, through pre-exposure of plants to a single or multiple stresses [28,61,62,63,64]. Although considered as a promising new strategy, there is still much to be explored around the mechanistic basis of adaptive stress memory, particularly its transgenerational effects in various crop species. The current study represents the first characterization of the impact of parental heat stress in the next generation in tetraploid wheat, providing new information on the underlying physiological and molecular mechanisms.

In cereal crops like wheat, grain production is heavily reliant on reproductive processes. One of the most prominent impacts of heat stress is the inhibition of plant photosynthesis capacity, due to thermo-damage and oxidative damage to chloroplasts, which ultimately affects dry matter accumulation and grain fill [11,65,66]. At the physiological level, such damage is often reflected as reduced chlorophyll content and inhibited photosynthetic rate in the leaves under heat stress [8,66]. In this study, parental exposure to heat stress had positive impacts on leaf chlorophyll content in the progeny. In DBA Aurora, the progeny heat stress treatment significantly reduced chlorophyll content in progeny groups from the control parents from 5DPA until 40DPA, and in progeny groups from the stressed parents from 15DPA until 40DPA. Interestingly, the stressed DBA Aurora progeny from the stress parents had significantly higher chlorophyll content than the stressed progeny from the control parents from 10DPA until 40DPA (except for 35DPA). The positive effect of parental stress was less evident in DBA Artemis, where the stressed DBA Artemis progeny from the stress parents only had significantly higher chlorophyll content than the stressed progeny from the control parents at 15DPA. The results suggested that heat stress treatment on the parents had contributed to less damage of the photosynthetic apparatus in the progeny undergoing heat stress, but the extent of beneficial impact varied depending on the genotype. Studies in other plant species have also demonstrated that leaf photosynthetic traits in the offspring are often affected by stressed parental environments [67,68]. In rice, a heavy metal stress treatment of the parent induced significantly higher chlorophyll content in the second generation progeny under the highest concentration of Hg^2+^ stress, demonstrating that progeny from the stressed plants developed heritable and enhanced stress tolerance [67]. In Arabidopsis, both chlorophyll content and O_2_ evolution readings were higher in seedlings from parents treated with iron deficiency, suggesting that the transgenerational effects of parental stress had improved the efficiency of the photosynthetic apparatus in the progeny [68]. The improvement of chlorophyll concentration was maintained for more than one generation, whereas the trait of higher O_2_ evolution was lost within two generations [68]. It is worth noting that the effects of parental stress might not always be positive, depending on the species. In a study using peanut, the progeny of the stressed parents actually showed decreased chlorophyll fluorescence parameters compared with progeny from the non-stressed parents [22]. Such changes were considered as a residual effect of the water-deficit stress imposed on the parents [22]. Here, within the same species, the effects of transgenerational heat stress were more notable in DBA Aurora than DBA Artemis. Therefore, research focusing on improving the photosynthetic performance of crops under the transgenerational influence of stress needs to be thoroughly investigated by using a wide range of germplasm. By doing so, breeding programs could target genotypes with improved photosynthetic parameters under the impact of transgenerational stress priming, and then utilize such genetic resources for selecting and breeding purposes.

The beneficial effects of transgenerational heat stress can also be observed for yield components and grain quality parameters measured in the current study. In DBA Aurora, control progeny from the stressed parents had significantly higher grain weight, grain number and biomass than the control progeny from the control parents, suggesting that parental stress priming could improve the yield for the next generation even without the reoccurrence of stress. Under progeny heat stress, progeny from the stressed parents not only had significantly higher grain weight, grain number and biomass than the stressed progeny from the control parents but these three yield traits of AuHH also showed no significant difference to AuCC. The results suggest that parental stress exposure could improve the tolerance of stressed progeny, to the point where its yield performance is comparable to unstressed plants from parents grown under optimal conditions. For grain quality traits, the transgenerational influence of heat stress was more pronounced in total starch content (TSC) than grain protein content (GPC). Studies in both tetraploid and hexaploid wheat have demonstrated that reproductive-stage heat stress generally leads to a higher GPC, often due to a shorter grain fill duration and smaller grain size, which results in more concentrated protein levels in the grain [5,8,69]. Here, both genotypes have exhibited the same trend for GPC, where a significant increase in GPC under heat stress was only found in progeny groups from the control parents, and stressed progeny from the stressed parents showed no significant difference to its control. Although a higher GPC is favorable (e.g., 14–16%) when it comes to durum grain quality assessment, heat-stress increased GPC is not always a preferred trait, as it is often associated with a concomitant reduction in grain weight and grain number, which means the overall grain protein yield has not improved [8]. Here, the stressed progeny groups from the stressed parents did not have higher GPC than its control, probably because they managed to maintain high grain weight and grain number, thus the protein deposition did not increase for individual grains. However, for grain TSC the beneficial influence of transgenerational stress was more pronounced. Under reproductive heat stress, TSC of cereal crops often decreases significantly due to inhibited photosynthesis that results in reduced starch biosynthesis and disrupted assimilate transport from photosynthetic organs to reproductive organs [8,70,71]. However, in the current study, both genotypes did not exhibit any significant difference in grain TSC between their stressed and control progeny groups from the stressed parents. For DBA Artemis, AtHH has significantly higher TSC than AtCC. For DBA Aurora, both AuHC and AuHH had significantly higher TSC than AuCH. These results suggest that the parental heat stress treatment influenced starch biosynthesis and transport processes in the progeny under heat stress, which could be attributed from less thermo-damage in the leaves as shown by the higher chlorophyll content. These results are in agreement with a previous study in bread wheat, where progeny of heat-primed parents had higher grain yield, higher photosynthetic rate and more active leaf dry matter remobilization when compared with progeny of non-primed plants [28]. It was also shown that the expression of photosynthesis related genes was up-regulated, and protein activities related to photosynthesis and energy production were increased, likely contributing to the improved thermo-tolerance in the progeny from the primed parents [28].

To efficiently utilize the positive impact of transgenerational stress in durum wheat breeding, we further explored the genetic and epigenetic changes at the molecular level that underpin the adaptive changes at the physiological and yield levels. Overall, we have identified a significant number of miRNAs and genes that were responsive to the progeny stress treatment and parental stress treatment factors. GO and KEGG pathway enrichment analyses have provided a global view of the functional pathways and biological processes that these DEMs and DEGs participate in under the influence of transgenerational heat stress. Our results contribute to the growing evidence that epigenetic regulators such as miRNAs are closely associated with stress memory passed on from parents to the next generation [24,25,26]. Particularly, under the influence of parental heat stress, the expression of heat stress responsive miRNAs can be primed to be down-regulated in the progeny, which would allow for the increased expression and activity of their target genes that encode protective proteins and enzymes, contributing to a higher level of thermo-tolerance under reoccurring heat stress. For example, in the current study, the expression of ata-miR528-5p was significantly lower in the leaf tissue of AuCH, AuHC and AuHH when compared with AuCC (Figure 6), indicating that miR528 was significantly down-regulated in response to heat stress in the progeny groups from the control parents and that miR528 expression was primed to be lower in the progeny groups from the stressed parents. Accordingly, the target of ata-miR528-5p, an F-box protein gene, showed the highest expression in the treatment AuHH group (Figure 7), suggesting that parental heat stress has induced higher expression of the F-box protein gene under reoccurring stress. As one of the largest protein families, F-box proteins play significant roles in plant adaptation to various abiotic stresses [72,73,74]. The F-box protein is a major subunit of the Skp1-Cullin-F-box (SCF) E3 ligase complex that regulates protein degradation and is characterized by a conserved motif of approximately 40 to 50 amino acids that functions as a protein-protein interaction site [72,75]. In previous studies in bread wheat, the expression of the wheat F-box protein gene *TaFBA1* was induced in response to oxidative and heat stress treatments in young seedlings [73,74]. The overexpression of *TaFBA1* enhanced the drought tolerance in transgenic plants through increased antioxidative capability, as shown by lower levels of reactive oxygen species (ROS), malondialdehyde content and cell membrane damage [73]. The overexpression of *TaFBA1* has also enhanced heat stress tolerance in transgenic tobacco [74]. An increase in photosynthesis and plant growth was found in transgenic plants compared to the wild type, as well as a decrease in the concentration of H_2_O_2_ and O_2_^.-^ [74]. Notably, under oxidative stress and heat stress, the chlorophyll content of transgenic plants was also significantly higher than the wild type [73,74]. This F-box protein interacts with the *Triticum aestivum* stress responsive protein 1 (TaASRP1), which also play key roles in heat stress response [73,74]. In the current study, an increased expression of an F-box protein gene through lowered miR528 expression due to transgenerational priming is beneficial to the stressed progeny plants. The interactions of F-box proteins with downstream proteins and substrates could contribute to higher antioxidative capability as seen in bread wheat, which would reduce cellular damage in the photosynthetic tissues, as shown by the increased chlorophyll content at the physiological level. Furthermore, this current study has shown that it is possible to induce miRNA-meidated transgenerational responses naturally, from the parents to the progeny, without the need to produce transgenics.

Two other miRNA-target modules, miR398-Cu/Zn SOD and miR5054-catalase could also contribute to an improved heat stress tolerance level in the progeny by promoting antioxidant capacity and ROS scavenging under the effects of transgenerational stress. Similar to ata-miR528-5p, the expression of osa-miR398a_L+1R-1 was significantly lower in the leaf tissue of AuCH, AuHC and AuHH when compared with AuCC (Figure 6). Moreover, the expression of bid-miR5054_L+2 was actually the lowest in the stressed progeny group from the stressed parents (AuHH). Significantly reduced miRNA expression under the impact of parental heat stress and progeny heat stress would lead to an increase of their target expression. Indeed, the Cu/Zn SOD gene targeted by osa-miR398a_L+1R-1, and the catalase (CAT) gene targeted by bid-miR5054_L+2, both had very high expression in the AuHH group (Figure 7). Both SOD and CAT are major antioxidant enzymes that function in ROS scavenging [64,76]. Under heat stress, the chloroplast thylakoid membrane is very sensitive to damage caused by ROS-induced lipid peroxidation reactions [77,78]. Disruption of thylakoid membranes inhibits the functionality of membrane-bound enzymes and electron carriers, which ultimately reduces the photosynthetic rate [77,78]. Antioxidant enzymes like SOD and CAT have ameliorating effects of ROS-induced damage, therefore the increase of SOD and CAT genes expressed via lowered miRNA expression would contribute to the maintenance of cellular homeostasis and higher membrane stability in the chloroplasts. Future research could further investigate the expression level and activities of these miRNA-target modules across multiple generations of heat stress, and determine their association with the activities of antioxidative enzymes and photosynthetic parameters in a range of germplasm with varying levels of heat stress tolerance.

In conclusion, our work demonstrates that parental heat-stress treatments at the reproductive stage can have a significant impact on how the next generation performs under heat stress on the molecular, physiological, and yield level. Particularly, changes in miRNA and mRNA gene expression on the transcriptome scale were pronounced, and key miRNA-mRNA modules related to ROS scavenging, signal transduction and photosynthesis play important roles in transgenerational stress response and can contribute to better adaptation to future stress encounters. The current study provides the first evidence of miRNA-mediated transgenerational heat stress response in durum wheat. The knowledge generated here provides new measures for enhancing crop resilience and yield stability in breeding of this economically important cereal.

## 4. Materials and Methods

### 4.1. Glasshouse Growing Conditions and Stress Treatments

Two South Australian durum wheat varieties, DBA Aurora and DBA Artemis, were used in this study. Seeds of two varieties were collected from the control group parents and post-anthesis heat stress group parents from a previous study [8]. Four seed groups were used: AuC (DBA Aurora, seeds from control group parents), AuH (DBA Aurora, seeds from heat stress group parents), AtC (DBA Artemis, seeds from control group parents) and AtH (DBA Artemis, seeds from heat stress group parents). Each seed group had two treatment groups—control and heat stress. In total, there were eight treatment groups: AuCC (DBA Aurora progeny treated with the control, originated from control group parents), AuCH (DBA Aurora progeny treated with heat stress, originated from control group parents), AuHC (DBA Aurora progeny treated with the control, originated from heat stress group parents), AuHH (DBA Aurora progeny treated with heat stress, originated from heat stress group parents), AtCC (DBA Artemis progeny treated with the control, originated from control group parents), AtCH (DBA Artemis progeny treated with heat stress, originated from control group parents), AtHC (DBA Artemis progeny treated with the control, originated from heat stress group parents), AtHH (DBA Artemis progeny treated with heat stress, originated from heat stress group parents).

Durum wheat plants were grown in pots (one plant per pot) in the glasshouse under controlled conditions as previously described [8]. The standard growing conditions were 22 °C/12 °C day/night temperature with a 12 h photoperiod. All plants were well watered from planting to harvest. Control group plants were grown under the standard growing temperature for the whole crop cycle. For the heat stress group, post-anthesis heat stress was applied at five timepoints [5, 15, 25, 35 and 45 days post anthesis (DPA)]. Plants were placed in a heat stress growth chamber at each timepoint under 37 °C/27 °C day/night temperature for 24 h, then they were moved back to the standard growing conditions as previously described [8].

### 4.2. Evaluation of Crop Performance and Statistical Analysis

Two leaf physiological traits (chlorophyll content and adaxial stomatal conductance) were measured on the flag leaf of the main stem every five days from flowering to 40DPA, consistently at midday [8,25,79]. Eight biological replicates per treatment groups were used for chlorophyll content measurement and six biological replicates were used for stomatal conductance measurement. One-way ANOVA was performed (GENSTAT 15th Edn, VSN International Ltd., Hemel Hempstead, UK) to determine statistical significance at *p* < 0.05 with the l.s.d. (least significance difference) for physiological trait measurements of all treatment groups within each genotype at each timepoint.

At maturity, harvest components including grain weight per plant, grain number per plant, plant biomass, plant height, tiller number per plant were measured (eight biological replicate per group). Harvest index and 1000-grain weight were calculated accordingly. Wholemeal durum flour was processed from the harvest grains of each biological replicate (four biological replicate per group) using an IKA A11 analytical mill. Grain quality traits including grain protein content (GPC), total starch content (TSC) and flour yellowness b* pigmentation (yellow-blue chromaticity) were determined with the wholemeal flour as previously described [8]. Briefly, GPC was measured using the Dumas combustible method (nitrogen content multiplied by the factor 5.7 at 11% moisture basis) with a Rapid N Elementar machine. TSC was determined using the Megazyme Total Starch Assay Kit (K-TSTA), according to the manufacture’s protocols. Flour yellowness b* was measured using a Konica Minolta Chroma Meter. One-way ANOVA was performed (GENSTAT 15th Edn, VSN International Ltd., Hemel Hempstead, UK) to determine statistical significance at *p* < 0.05 with the least significant difference (LSD) value for yield components and grain quality trait measurements of all treatment groups within each genotype.

### 4.3. High-Throughput Small RNA Sequencing, Transcriptome Sequencing and Degradome Sequencing

Flag leaf and developing grains of the main tiller were sampled from four biological replicates per treatment group at 5 DPA. Total RNA was extracted from the samples using the Tri reagent (Sigma-Aldrich, North Ryde, Australia). Extracted RNA was treated with TURBO DNase (ThermoFisher Scientific, Waltham, MA, USA) to remove any genomic DNA. RNA quality, concentration and integrity measured with gel electrophoresis, NanoDrop spectrophotometer and the Agilent Bioanalyzer. Equal amount of RNA from the biological replicates were pooled prior to sequencing library construction. DBA Aurora samples were used for next-generation sequencing analysis. In total, there were eight small RNA libraries, eight mRNA transcriptome libraries and eight mRNA degradome libraries (four DBA Aurora treatment groups by two tissue types).

Small RNA libraries were constructed as previously described using the NEBNext Multiplex Small RNA Library Prep Kit and multiplex primers set [25,26,48,49]. Sequencing of the small RNA libraries were performed on an Illumina HiSeq 2500 at LC-Bio (Hangzhou, China). The sequencing datasets generated in the current study (small RNA sequencing, transcriptome sequencing and degradome sequencing) have been deposited into the NCBI GEO database (accession number GSE171303). Bioinformatics analysis of the sRNA sequencing data was performed using the ACGT101-miR program as previously described [25,26,48,49]. Briefly, conserved miRNAs were identified by BLAST searching clean filtered sRNA reads against plant miRNAs registered in the miRBase (Release 22.1) [26]. The remaining unmatched sRNA reads were used for novel miRNA identification, where miRNA secondary hairpin structures were analyzed using the RNAfold tool [26,80]. All identified miRNAs were categorized into five miRNA groups (Group 1 to Group 5) as previously described [26]. Group 1 to Group 4 include different types of conserved miRNAs identified in this study, and Group 5 contains all the novel miRNAs that are unique to durum wheat. The normalized reads count of miRNAs in each library was used for miRNA differential expression analysis. Chi-square (2 × 2) was used to identify differentially expressed miRNAs (DEMs) subject to each treatment factor at *p* < 0.05. DEM log2 (fold-change) was calculated for each paired-comparison made between two treatment groups.

Transcriptome libraries were constructed using the Illumina mRNA-Seq sample preparation kit as previously described [25,48,49]. RNA-seq was performed on an Illumina NovaSeq 6000 at LC-Bio (Hangzhou, China). Bioinformatics analysis was performed as previously described [25,48,49]. Briefly, low quality reads were removed prior to sequence assembly. Filtered clean reads were aligned to the durum wheat reference genome using the HISAT package. The aligned reads of each RNA-seq library were assembled and the abundance of transcripts were obtained using the StringTie tool. Normalized relative abundance of transcripts was expressed as FPKM (Fragments Per Kilobase Million). Chi-square (2 × 2) was used to identify differentially expressed genes (DEGs) subject to each treatment factor at *p* < 0.05. DEG log2 (fold-change) was calculated for each paired-comparison made between two treatment groups.

For degradome sequencing, magnetic beads-enriched mRNA was used for library construction [25,48,49]. Enriched mRNA samples were mixed with biotinylated random primers, and 5’ adaptors were ligated. First-strand cDNA reverse transcription was performed with the ligated mRNA samples. After PCR amplification, cDNA libraries were sequenced on an Illumina Hiseq2500 at LC-BIO (Hangzhou, China). Bioinformatics analysis was performed using the ACGT101-DEG program and the CleaveLand package (V4.0) as previously described [25,48,49]. Target mRNA transcripts of novel and conserved durum wheat miRNAs were identified. Reads abundance of target transcripts were normalized to transcripts per billion (TPB).

### 4.4. Intergrated Omics Analysis and Functional Enrichment Analyses of Genes

Gene Ontology enrichment analysis and KEGG pathway enrichment analysis were performed for DEGs and mRNA targets of DEGs as previously described. Briefly, GO terms of DEGs and mRNA target transcripts were annotated in three GO categories: biological process (BP), cellular component (CC) and molecular function (MF). Enrichment analysis of KEGG pathways associated with DEGs and mRNA targets was performed to identify the significant regulatory pathways that miRNA-target modules and stress responsive genes are associated with. The integrated omics analysis of sRNA sequencing, mRNA transcriptome sequencing and degradome sequencing was conducted to discover and validate significant miRNA-mRNA pairs with antagonistic regulatory patterns. First, miRNA and target mRNA pairs were validated based on the three types of sequencing datasets (i.e., miRNA candidates confirmed by valid sRNA sequencing reads, mRNA candidates confirmed by valid RNA sequencing reads and miRNA targeted-mRNA pairing confirmed by valid degradome sequencing reads). Second, from all the confirmed miRNA-target mRNA pairs, those where both miRNA and the targeted gene had shown significant differential expression profile were identified. Last, from all the miRNA-target mRNA pairs with significant differential expression, those were miRNA and its target gene had shown an antagonistic regulatory pattern were identified (i.e., significantly down-regulated miRNA corresponding to significantly up-regulated target gene expression, or significantly up-regulated miRNA corresponding to significantly down-regulated target gene expression). Enrichment analyses of GO terms and KEGG pathway were performed for these significant antagonistic miRNA-target mRNA pairs.

### 4.5. qPCR Profiling of DEMs and DEGs

Seven stress-responsive miRNAs and 11 target genes were chosen for qPCR expression analysis. The MystiCq microRNA cDNA Synthesis Mix kit (Sigma-Aldrich, North Ryde, Australia) was used for cDNA synthesis as previously described. qPCR experiment was performed using the PowerUp SYBR Green Master Mix on a ViiA7 Real-Time PCR machine as previously described. The durum wheat GAPDH gene was used as the reference gene to determine the relative expression level of miRNAs and their target genes. Relative gene expression was shown as mean ± SE (three biological replicates). One-way ANOVA analysis was performed to identify statistical significance across treatment groups (*p* < 0.05). Different letters (a, b, c) represent significant statistical difference across treatment groups based on the l.s.d value. Primers used in this study are shown in Appendix A.

## Figures and Tables

**Figure 1 ijms-22-05532-f001:**
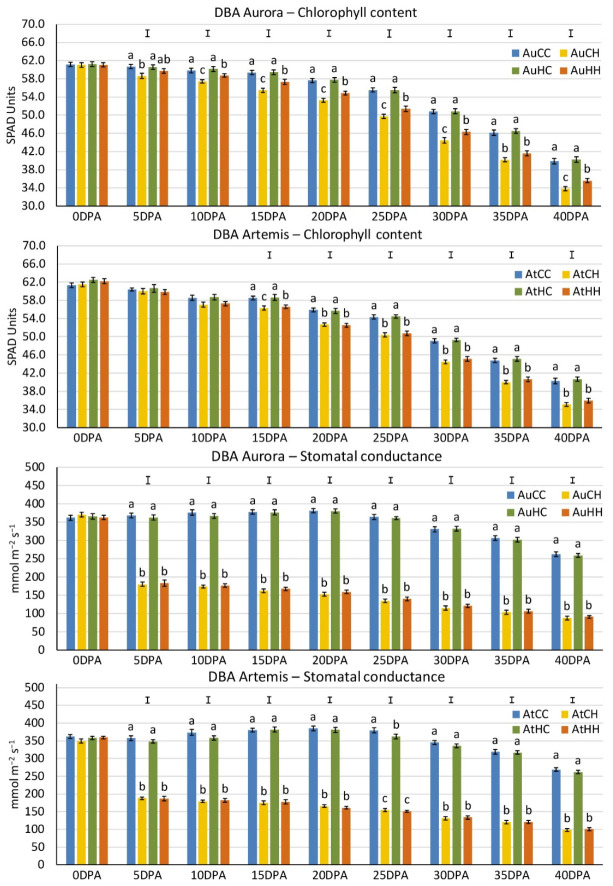
Flag leaf chlorophyll content and stomatal conductance (adaxial) measured from flowering to 40 days post-anthesis (DPA) in two durum wheat varieties. Means are shown with standard error (*n* = 8 for chlorophyll content, *n* = 4 for stomatal conductance). The capped lines show the LSD (Least Significant Difference) values at *p* = 0.05 for measurements taken at each time-point. The letters (a, b, c) indicate significant statistical difference (*p* < 0.05) across treatment groups at each time-point. The treatment groups are: AuCC (DBA Aurora progeny treated with the control, originated from control group parents), AuCH (DBA Aurora progeny treated with heat stress, originated from control group parents), AuHC (DBA Aurora progeny treated with the control, originated from heat stress group parents), AuHH (DBA Aurora progeny treated with heat stress, originated from heat stress group parents), AtCC (DBA Artemis progeny treated with the control, originated from control group parents), AtCH (DBA Artemis progeny treated with heat stress, originated from control group parents), AtHC (DBA Artemis progeny treated with the control, originated from heat stress group parents), AtHH (DBA Artemis progeny treated with heat stress, originated from heat stress group parents).

**Figure 2 ijms-22-05532-f002:**
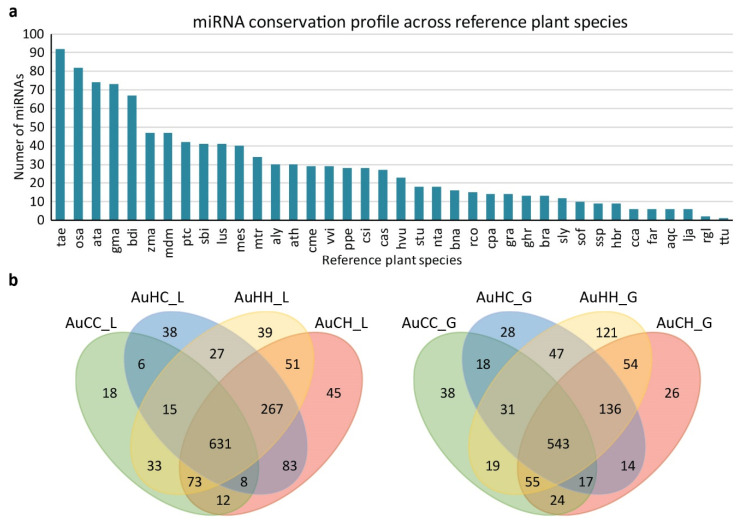
The miRNA conservation profile across reference plant species (**a**) and the distribution of miRNAs across treatment groups in DBA Aurora (**b**). The treatment groups are: AuCC (DBA Aurora progeny treated with the control, originated from control group parents), AuCH (DBA Aurora progeny treated with heat stress, originated from control group parents), AuHC (DBA Aurora progeny treated with the control, originated from heat stress group parents), AuHH (DBA Aurora progeny treated with heat stress, originated from heat stress group parents). _L indicates flag leaf library and _G indicates developing grains library. The abbreviations of plant reference species are: tae, *Triticum aestivum*. osa, *Oryza sativa*. ata, *Aegilops tauschii*. gma, *Glycine max*. bdi, *Brachypodium distachyon*. zma, *Zea mays*. mdm, *Malus domestica*. ptc, *Populus trichocarpa*. sbi, *Sorghum bicolor*. lus, *Linum usitatissimum*. mes, *Manihot esculenta*. mtr, *Medicago truncatula*. aly, *Arabidopsis lyrata*. ath, *Arabidopsis thaliana*. cme, *Cucumis melo*. vvi, *Vitis vinifera*. ppe, *Prunus persica*. csi, *Citrus sinensis*. cas, *Camelina sativa*. hvu, *Hordeum vulgare*. stu, *Solanum tuberosum*. nta, *Nicotiana tabacum*. bna, *Brassica napus*. rco, *Ricinus communis*. cpa, *Carica papaya*. gra, *Gossypium raimondii*. ghr, *Gossypium hirsutum*. bra, *Brassica rapa*. sly, *Solanum lycopersicum*. sof, *Saccharum officinarum*. ssp, *Saccharum ssp*. hbr, *Hevea brasiliensis*. cca, *Cynara cardunculus*. far, *Festuca arundinacea*. aqc, *Aquilegia caerulea*. lja, *Lotus japonicas*. rgl, *Rehmannia glutinosa*. ttu, *Triticum turgidum*.

**Figure 3 ijms-22-05532-f003:**
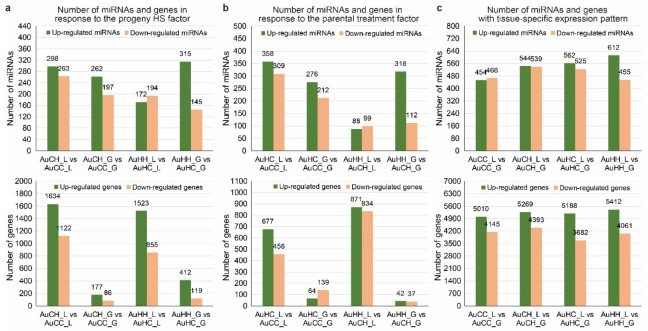
The number of up-regulated and down-regulated miRNAs and genes subject to different factors: (**a**) progeny heat stress treatment, (**b**) parental heat stress treatment, (**c**) tissue type. The treatment groups are: AuCC (DBA Aurora progeny treated with the control, originated from control group parents), AuCH (DBA Aurora progeny treated with heat stress, originated from control group parents), AuHC (DBA Aurora progeny treated with the control, originated from heat stress group parents), AuHH (DBA Aurora progeny treated with heat stress, originated from heat stress group parents). _L indicates flag leaf library and _G indicates developing grains library.

**Figure 4 ijms-22-05532-f004:**
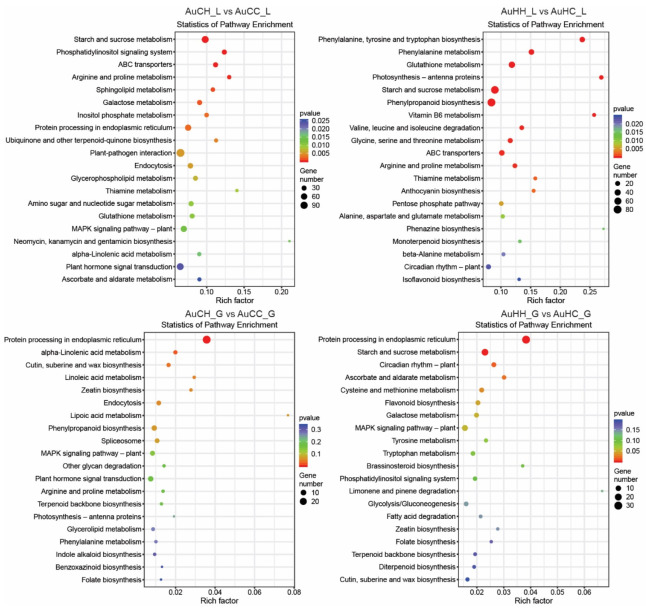
Scatter plots showing the top enriched KEGG pathways associated with progeny heat stress-responsive genes in different pair-wise comparisons. The treatment groups are: AuCC (DBA Aurora progeny treated with the control, originated from control group parents), AuCH (DBA Aurora progeny treated with heat stress, originated from control group parents), AuHC (DBA Aurora progeny treated with the control, originated from heat stress group parents), AuHH (DBA Aurora progeny treated with heat stress, originated from heat stress group parents). _L indicates flag leaf library and _G indicates developing grains library. The size of the dots is representative of the number of genes enriched under each KEGG pathway. The rich factor indicates the degree of enrichment, which was calculated based on the ratio of heat stress-responsive genes:all genes that can be annotated to a specific KEGG pathway.

**Figure 5 ijms-22-05532-f005:**
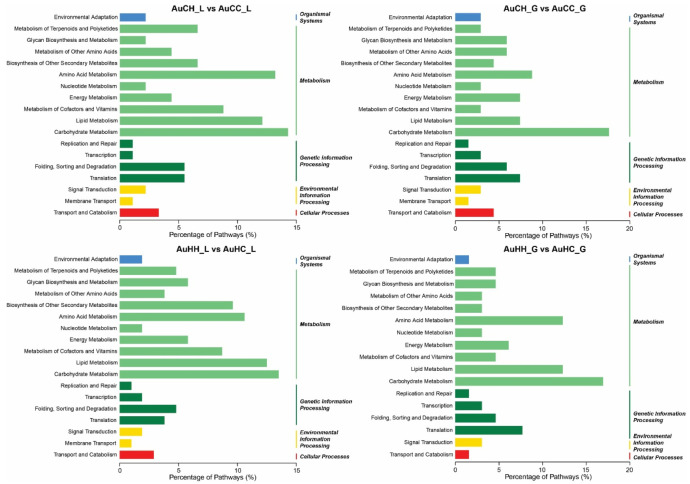
KEGG pathway classification of progeny heat stress-responsive miRNA-mRNA regulatory modules in different pair-wise comparisons. The treatment groups are: AuCC (DBA Aurora progeny treated with the control, originated from control group parents), AuCH (DBA Aurora progeny treated with heat stress, originated from control group parents), AuHC (DBA Aurora progeny treated with the control, originated from heat stress group parents), AuHH (DBA Aurora progeny treated with heat stress, originated from heat stress group parents). _L indicates flag leaf library and _G indicates developing grains library.

**Figure 6 ijms-22-05532-f006:**
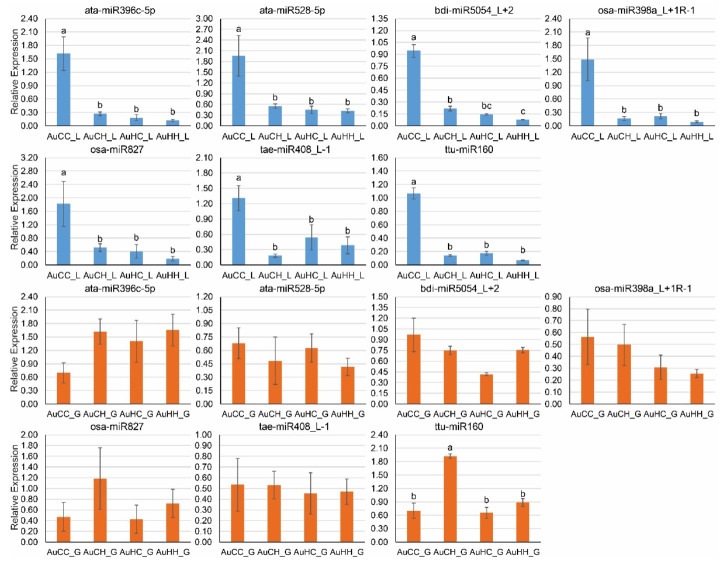
qPCR expression profiles of seven selected stress-responsive miRNAs in DBA Aurora. Relative miRNA expression was calculated using GAPDH as the housekeeping gene. Data are presented as the mean ± standard error (SE) (*n* = 3). The treatment groups are: AuCC (DBA Aurora progeny treated with the control, originated from control group parents), AuCH (DBA Aurora progeny treated with heat stress, originated from control group parents), AuHC (DBA Aurora progeny treated with the control, originated from heat stress group parents), AuHH (DBA Aurora progeny treated with heat stress, originated from heat stress group parents). _L indicates flag leaf samples (highlighted in blue) and _G indicates developing grains samples (highlighted in orange). One-way ANOVA analysis was performed to identify statistical significance across treatment groups (*p* < 0.05). Different letters (a, b, c) represent significant statistical difference across treatment groups based on the l.s.d value.

**Figure 7 ijms-22-05532-f007:**
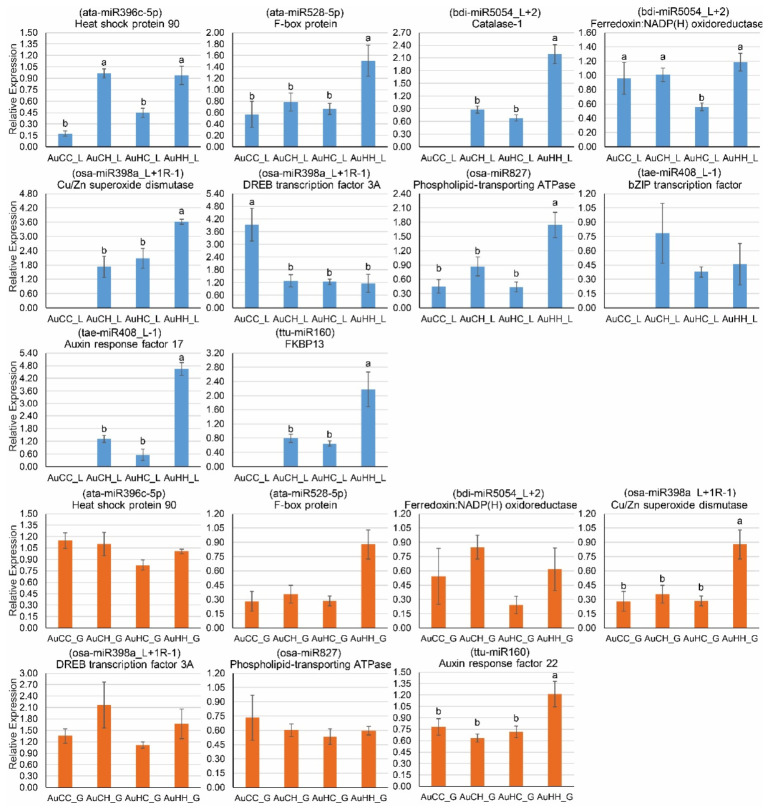
qPCR expression profile of 11 selected stress-responsive target genes in DBA Aurora. Relative miRNA expression was calculated using GAPDH as the housekeeping gene. Data are presented as the mean ± standard error (SE) (*n* = 3). The treatment groups are: AuCC (DBA Aurora progeny treated with the control, originated from control group parents), AuCH (DBA Aurora progeny treated with heat stress, originated from control group parents), AuHC (DBA Aurora progeny treated with the control, originated from heat stress group parents), AuHH (DBA Aurora progeny treated with heat stress, originated from heat stress group parents). _L indicates flag leaf samples (highlighted in blue) and _G indicates developing grains samples (highlighted in orange). One-way ANOVA analysis was performed to identify statistical significance across treatment groups (*p* < 0.05). Different letters (a, b, c) represent significant statistical difference across treatment groups based on the l.s.d value. FKBP13: peptidyl-prolyl cis-trans isomerase FKBP13, chloroplastic-like. The auxin response factor 22 (target of ttu-miR160) was not expressed in the flag leaf tissue. The catalase-1 gene (target of bid-miR5054_L+2), bZIP transcription factor (target of tae-miR408_L-1), auxin response factor 17 (target of tae-miR408_L-1) and FKBP13 (target of ttu-miR160) were not expressed in the developing grains.

**Table 1 ijms-22-05532-t001:** Progeny yield performance under the effects of transgenerational and progeny heat stress. Means are shown with standard error (*n* = 8). The letters (a, b, c) indicate significant statistical difference (*p* < 0.05) across treatment groups within each variety. The treatment groups are: AuCC (DBA Aurora progeny treated with the control, originated from control group parents), AuCH (DBA Aurora progeny treated with heat stress, originated from control group parents), AuHC (DBA Aurora progeny treated with the control, originated from heat stress group parents), AuHH (DBA Aurora progeny treated with heat stress, originated from heat stress group parents), AtCC (DBA Artemis progeny treated with the control, originated from control group parents), AtCH (DBA Artemis progeny treated with heat stress, originated from control group parents), AtHC (DBA Artemis progeny treated with the control, originated from heat stress group parents), AtHH (DBA Artemis progeny treated with heat stress, originated from heat stress group parents).

	Grain Weight (g)	Grain Number	Biomass (g)	Harvest Index	1000-Grain Weight (g)
AtCC	12.82 ± 0.31 a	330.38 ± 6.72 a	25.92 ± 0.36 a	0.496 ± 0.017 a	38.98 ± 1.48
AtCH	10.36 ± 0.72 b	273.43 ± 13.29 bc	23.56 ± 0.52 b	0.437 ± 0.021 b	37.99 ± 2.00
AtHC	12.18 ± 0.27 a	306.13 ± 5.54 ab	25.73 ± 0.37 a	0.473 ± 0.005 ab	39.84 ± 0.85
AtHH	10.50 ± 0.49 b	271.50 ± 18.50 b	23.52 ± 0.71 b	0.446 ± 0.013 b	39.40 ± 2.01
F pr.	0.001	0.005	0.002	0.037	0.879
LSD	1.35	35.27	1.48	0.043	n.a.
AuCC	13.67 ± 0.23 b	283.00 ± 6.34 b	24.65 ± 0.23 b	0.555 ± 0.008 ab	48.35 ± 0.54
AuCH	11.23 ± 0.43 c	250.50 ± 12.45 c	21.16 ± 0.47 c	0.530 ± 0.013 bc	45.12 ± 1.11
AuHC	14.80 ± 0.18 a	325.25 ± 8.03 a	26.38 ± 0.30 a	0.561 ± 0.002 a	45.66 ± 1.03
AuHH	12.94 ± 0.43 b	286.88 ± 15.19 b	24.69 ± 0.54 b	0.523 ± 0.008 c	45.52 ± 1.30
F pr.	<0.001	<0.001	<0.001	0.013	0.132
LSD	0.97	32.07	1.18	0.026	n.a.

**Table 2 ijms-22-05532-t002:** Progeny grain quality and morphological performance under the effects of transgenerational and progeny heat stress. Means are shown with standard error (*n* = 8 for morphological traits, *n* = 8 for grain quality traits). The letters (a, b, c) indicate significant statistical difference (*p* < 0.05) across treatment groups within each variety. The treatment groups are: AuCC (DBA Aurora progeny treated with the control, originated from control group parents), AuCH (DBA Aurora progeny treated with heat stress, originated from control group parents), AuHC (DBA Aurora progeny treated with the control, originated from heat stress group parents), AuHH (DBA Aurora progeny treated with heat stress, originated from heat stress group parents), AtCC (DBA Artemis progeny treated with the control, originated from control group parents), AtCH (DBA Artemis progeny treated with heat stress, originated from control group parents), AtHC (DBA Artemis progeny treated with the control, originated from heat stress group parents), AtHH (DBA Artemis progeny treated with heat stress, originated from heat stress group parents).

	Grain Protein Content (GPC%)	Total Starch Content (TSC%)	Flour Yellowness (*b)	Plant Height (cm)	Tiller Number	Main Spike Length (cm)
AtCC	11.66 ± 0.31 b	52.03 ± 6.72 c	22.89 ± 0.23	70.40 ± 1.00	5.50 ± 0.27	8.79 ± 0.07 b
AtCH	14.21 ± 1.06 a	55.10 ± 13.29 bc	22.99 ± 0.26	71.46 ± 1.32	5.14 ± 0.14	8.83 ± 0.14 b
AtHC	11.72 ± 0.26 b	61.18 ± 1.61 a	23.09 ± 0.31	70.39 ± 1.23	5.50 ± 0.19	9.09 ± 0.10 ab
AtHH	13.39 ± 0.33 ab	58.65 ± 1.31 ab	23.35 ± 0.24	73.30 ± 1.28	5.38 ± 0.32	9.35 ± 0.15 a
F pr.	0.024	0.011	0.644	0.286	0.726	0.007
LSD	1.826	5.125	n.a.	n.a.	n.a.	0.34
AuCC	11.90 ± 0.23 b	54.03 ± 1.32 ab	21.24 ± 0.36	71.04 ± 1.56	5.50 ± 0.19	8.80 ± 0.13
AuCH	13.18 ± 0.58 a	51.82 ± 1.16 b	21.27 ± 0.30	74.21 ± 0.75	5.25 ± 0.37	8.93 ± 0.13
AuHC	11.82 ± 0.22 b	56.13 ± 0.91 a	21.93 ± 0.28	75.78 ± 1.02	5.63 ± 0.26	9.01 ± 0.17
AuHH	12.85 ± 0.28 ab	56.41 ± 1.06 a	21.99 ± 0.29	73.43 ± 1.33	6.00 ± 0.38	9.01 ± 0.10
F pr.	0.045	0.045	0.210	0.066	0.397	0.654
LSD	1.102	3.456	n.a.	n.a.	n.a.	n.a.

## Data Availability

The small RNA, transcriptome and degradome sequencing datasets generated in the current study have been deposited in the NCBI GEO database under the accession number GSE171303.

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
