# Peer review of "Small RNA, Transcriptome and Degradome Analysis of the Transgenerational Heat Stress Response Network in Durum Wheat"

_ijms, 2021, doi:10.3390/ijms22115532_

Round 1
Reviewer 1 Report
Dear Authors,
In general, I have no major comments on the manuscript. Please only add authority to the botanical name in line 28. Also, please reword the last paragraph of the introduction so that it clearly states the purpose of the study rather than being a summary of the results.
Regards,
M.
Author Response
Reviewer 1
Dear Authors,
In general, I have no major comments on the manuscript. Please only add authority to the botanical name in line 28. Also, please reword the last paragraph of the introduction so that it clearly states the purpose of the study rather than being a summary of the results.
Regards,
M.
Response: We thank Reviewer 1 for reviewing our manuscript. We have only kept the botanical name in line 28 and removed the other (line 33). We have revised the last paragraph of the introduction to state the purpose of the study (lines 115-121).

Reviewer 2 Report
It is a well-written manuscript, including well-performed experiments and an excessive dataset. Genetic variation on the effect of parental exposure to heat stress on progeny response was addressed including a range of investigated aspects.
Only few minor comments:
(1) Line 59: ROS are not protective metabolites (in fact they are the opposite)
(2) The priming effect (Line 529) and ROS-scavenging role of CAT and SOD (Line 675) are also dealt in a very recent study (Chen et al., 2021 Postharvest Biol Technol 172, 111376)
(3) Stomatal conductance measurement (Line 725): why adaxial (upper) leaf side? Which time of the day did you conduct the measurement?
Author Response
Reviewer 2
It is a well-written manuscript, including well-performed experiments and an excessive dataset. Genetic variation on the effect of parental exposure to heat stress on progeny response was addressed including a range of investigated aspects.
Response: We thank Reviewer 2 for acknowledging the value of our work and providing the valuable comments. A point-by-point response is listed below.
Only few minor comments:
(1) Line 59: ROS are not protective metabolites (in fact they are the opposite)
Response: We have corrected this to “ROS scavengers” (line 59), which are protective metabolites.
(2) The priming effect (Line 529) and ROS-scavenging role of CAT and SOD (Line 675) are also dealt in a very recent study (Chen et al., 2021 Postharvest Biol Technol 172, 111376)
Response: The reference has been added (line 600 & line 743).
(3) Stomatal conductance measurement (Line 725): why adaxial (upper) leaf side? Which time of the day did you conduct the measurement?
Response: We chose to measure the stomatal conductance on the adaxial surface to be consistent with previous studies in wheat (e.g. DOI: 10.1111/ppl.12245; DOI: 10.1038/s41598-019-49871-x; DOI: 10.1038/s41598-021-83074-7). References have been added here (line 795). Stomatal conductance was measured consistently at midday (line 795).